# Exogenous Ketone Supplementation and Ketogenic Diets for Exercise: Considering the Effect on Skeletal Muscle Metabolism

**DOI:** 10.3390/nu15194228

**Published:** 2023-09-30

**Authors:** Hannah Khouri, John R. Ussher, Céline Aguer

**Affiliations:** 1Department of Biochemistry, Microbiology and Immunology, Faculty of Medicine, University of Ottawa, Ottawa, ON K1N 6N5, Canada; hkhou006@uottawa.ca; 2Institut du Savoir Montfort, Hôpital Montfort, Ottawa, ON K1K 0T2, Canada; 3Faculty of Pharmacy and Pharmaceutical Sciences, University of Alberta, Edmonton, AB T6G 2H5, Canada; jussher@ualberta.ca; 4Department of Physiology, Faculty of Medicine and Health Sciences, McGill University–Campus Outaouais, Gatineau, QC J8V 3T4, Canada

**Keywords:** ketone bodies, ketogenic diet, ketone supplements, skeletal muscle, exercise, exercise performance, metabolism

## Abstract

In recent years, ketogenic diets and ketone supplements have increased in popularity, particularly as a mechanism to improve exercise performance by modifying energetics. Since the skeletal muscle is a major metabolic and locomotory organ, it is important to take it into consideration when considering the effect of a dietary intervention, and the impact of physical activity on the body. The goal of this review is to summarize what is currently known and what still needs to be investigated concerning the relationship between ketone body metabolism and exercise, specifically in the skeletal muscle. Overall, it is clear that increased exposure to ketone bodies in combination with exercise can modify skeletal muscle metabolism, but whether this effect is beneficial or detrimental remains unclear and needs to be further interrogated before ketogenic diets or exogenous ketone supplementation can be recommended.

## 1. Introduction

Ketone bodies (hereinafter referred to as ketones) are produced by the liver during periods of fasting or low carbohydrate availability to provide an alternative source of energy to glucose and fatty acids [1,2]. Specifically, the ketones β-hydroxybutyrate (βOHB) and acetoacetate are synthesized from acetyl-CoA derived from hepatic fatty acid oxidation, then they are exported to extra-hepatic tissues primarily for oxidation. The brain is a major consumer of ketones during fasting, but other tissues such as the skeletal muscle (hereinafter referred to as muscle) can also oxidize ketones for energy. This is advantageous because it limits muscle proteolysis to generate amino acids, which can feed hepatic glucose production. Importantly, the rate-limiting enzyme of ketogenesis is HMGCS2 (3-hydroxymethylglutaryl-CoA synthase 2), and the enzymes BDH (βOHB dehydrogenase), SCOT (succinyl-CoA:3-oxoacid-CoA transferase; gene name *Oxct1*) and ACAT (acetoacetyl CoA thiolase) sequentially catalyze the series of steps in ketone body oxidation, where βOHB can be reversibly converted to acetoacetate by BDH. Of note, βOHB exists as two different enantiomers, D-βOHB and L-βOHB [3]. The literature suggests that D-βOHB is involved in ketogenesis and ketolysis (and is therefore oxidized to produce adenosine triphosphate (ATP)), whereas the role of L-βOHB and how it is produced remains unclear [4].

In recent years, ketogenic diets (hereinafter referred to as keto diets) have increased in popularity. The diet consists of very low levels of carbohydrates in combination with high levels of fat and moderate levels of protein. As a result, the body enters a state of ketosis, where it becomes dependent on burning fat and ultimately synthesizing ketones for energy [5]. The levels of ketones in the organism can also be increased exogenously without the need to limit carbohydrate intake via supplementation with ketone esters. Ketone esters are typically consumed in a beverage and have been previously proven to enter the bloodstream to temporarily increase circulating levels of ketones [6].

It has been proposed that a keto diet or exogenous ketone supplementation may be beneficial with respect to exercise [7,8]. This hypothesis is based mainly on the presumption that increased ketolysis could alter muscle fuel selection during exercise by favoring lipid metabolism over carbohydrate metabolism. Our bodies also have a limited capacity to store carbohydrates as glycogen, relative to the capacity to generate a lipid reserve. Therefore, increasing fat utilization may be a mechanism to increase the amount of energy available during exercise, especially in endurance activities. As such, a variety of studies (which will be highlighted in this review) have investigated the potential effect of increased ketolysis on exercise performance and the associated metabolic adaptations (e.g., muscle glycogen). When considering the impact of a keto diet on metabolism and exercise, the muscle is an important tissue to study, as it accounts for the majority of post-prandial glucose uptake [9]. The muscle is equally a critical tissue to consider when investigating factors related to physical activity, since it is required for locomotion [10]. Therefore, this review will summarize what is known to date regarding the relationship between increased ketolysis and exercise, with an emphasis on their effects on muscle metabolism.

## 2. Effect of Exercise on Muscle Ketone Body Metabolism

Parameters linked to ketone body metabolism include muscle ketolytic gene expression, enzyme activity and rates of ketolysis. Existing literature regarding the effect of exercise and increased exposure to ketone bodies on these factors is outlined below. To begin, the reported effect of a keto diet and exercise on the muscle expression of enzymes involved in ketone body metabolism is variable. A 12-week keto diet (88% fat, 11% protein and 1% carbohydrates) suppressed *Oxct1* mRNA expression in the gastrocnemius muscle of male mice (relative to a standard diet), but the combination of the keto diet and eight weeks of treadmill exercise had no additional effect on *Oxct1* expression [11], suggesting that treadmill training does not counteract the negative effect of a keto diet on *Oxct1* expression, at least in mice. However, the same mice exposed to both a keto diet and exercise showed an increased expression of *Hmgcs2* mRNA in the gastrocnemius muscle. Meanwhile, exercise alone or the keto diet alone did not alter the expression *Hmgcs2*. Therefore, the combination of exercise training and a keto diet seems to have a positive effect on the expression of enzymes involved in ketogenesis in mice, yet it remains unknown whether the muscle can support ketogenesis even if *Hmgcs2* is expressed. Next, an eight-week keto diet (76.1% fat, 8.9% protein and 3.5% carbohydrates) combined or not with exercise did not alter *Oxct1* mRNA expression in the gastrocnemius or soleus muscle of male mice [8]. Yet, the mRNA expression of *Bdh* was downregulated by the keto diet in the gastrocnemius muscle (mixed muscle) but upregulated in the soleus (oxidative muscle). Although this effect was not dependent on exercise, it highlights the potential variability in gene expression in different muscle types in response to a keto diet or exercise. Further, the shorter-term keto diet may explain why different results were reported as compared to the 12-week keto diet described above. Since medium-chain triacylglycerols (TAGs) are believed to be more ketogenic due to rapid uptake into the liver, a third study further separated an 8-week regimen into either a keto diet with long-chain or medium-chain TAGs, with regular swimming exercise. In general, male rats who performed the exercise regimen had higher SCOT protein expression across all diets [12]. Further, SCOT protein expression was increased in the epitrochlearis muscle with both keto diets relative to a control diet, but to a greater extent with the medium-chain TAGs. This conclusion suggests that the composition of a keto diet is an important variable to consider. Other studies took a different approach and measured the effect of acute exercise on ketolytic gene expression independently of diet. For example, a single bout of treadmill running did not alter *Bdh*, *Oxct1* or *Acat* expression in mouse quadriceps muscle [13].

In addition to their expression, a small selection of studies investigated the impact of exercise on the activity of enzymes involved in ketolysis. Following a 10-week treadmill running program, SCOT activity was increased in the gastrocnemius muscle (mixed muscle) of male rats [14]. A comparable study reported the same results, but they also measured the activity of BDH and ACAT, which were also increased in the gastrocnemius muscle as a result of the exercise regimen [15]. On the other hand, following 15 weeks of treadmill running in male rats, SCOT activity was increased in the diaphragm (mixed muscle), but no differences were reported in the intercostal muscle (glycolytic muscle) [16]. It should be noted that these are respiratory muscles, not muscles involved in movement during exercise. All together, these studies suggest that exercise increases SCOT activity, but there may be variability between muscle types, potentially with a less important effect observed in glycolytic muscles. Interestingly, it is also possible that metabolic diseases impact the effect of exercise on SCOT activity, as a 10-week exercise program in diabetic rats increased gastrocnemius SCOT activity to a greater extent than in rats without diabetes (when measured relative to their sedentary counterparts) [14]. The greater increase is likely because sedentary diabetic rats had lower SCOT activity than the rats without diabetes, but regardless, the observation implies that exercise may protect against decreased muscle ketolysis induced by diabetes (which may be connected to diabetic hyperketonemia).

Lastly, there have also been reports on the impact of exercise on the rate of ketone body utilization in the muscle. For example, 12 weeks of treadmill running increased oxidation of acetoacetate and D,L-βOHB in isolated gastrocnemius muscle from male rats [15]. Similarly, the uptake of acetoacetate alone and a combination of acetoacetate and βOHB was increased in perfused hindlimb muscles of trained relative to untrained male rats (although no effect was reported for the uptake of βOHB alone) [17]. Contrarily, a single bout of treadmill running did not alter maximal acetoacetate or βOHB supported respiration in permeabilized fibers from the gastrocnemius of male rats [18]. It is possible that a prolonged exercise program is necessary to induce alterations in ketone body oxidation or ketolytic gene expression and activity, and the length of a keto diet or exercise program should be taken into consideration.

Other studies have measured the levels of ketones in the muscle following exercise, although it is unclear whether increased muscle ketones are indicative of increased ketone body uptake and oxidation or decreased muscle ketolysis. For example, in male endurance athletes who consumed a beverage supplemented with ketone esters during an exercise test, the levels of D-βOHB in the muscle were higher 1 h post-exercise (relative to athletes who consumed a beverage without ketone esters) [7]. However, the levels were also higher pre-exercise, which implies that the D-βOHB may have been minimally oxidized by the muscle during the activity. As another example, in in vitro C2C12 mouse muscle cells exposed to forskolin (a cAMP pathway activator to mimic exercise), levels of βOHB were higher in cell lysates and culture media than in control cells [19]. To the best of our knowledge, this is the only study that suggests that the muscle may produce βOHB. Since the study cited above concluded that *Hmgsc2* is expressed in the gastrocnemius muscle of male mice, it supports the possibility that the muscle is capable of undergoing ketogenesis, a hypothesis that should be explored in more depth in the future.

To summarize, it appears that exercise training increases the expression of enzymes involved in ketone body metabolism, as well as their activity and rates of ketolysis in the muscle. However, variables such as the length and composition of the keto diet, and the muscle type in question should be considered. Additionally, it should be noted that the majority of studies on ketones, exercise and the muscle are conducted exclusively in male rodents. However, although only measured in the plasma and not the muscle, it was reported that the levels of serum acetoacetate and βOHB were higher in female than male mice following a bout of endurance exercise [13]. This conclusion highlights the need to introduce biological sex as a variable in experiments similar to the ones described here. Further, almost all of the studies presented in this section were performed in rodents or rodent cell lines. As such, no conclusions can be drawn regarding the effect of ketone bodies on ketolytic gene expression and enzyme activity in human muscle, a limitation that should be investigated in future experiments.

## 3. Muscle PDK4 Activation Following a Keto Diet and Exercise

Pyruvate dehydrogenase (PDH) is a mitochondrial enzyme complex that converts pyruvate into acetyl-CoA, thereby controlling the entry of glucose into the Krebs cycle. In the muscle, PDH is inhibited by phosphorylation via pyruvate dehydrogenase kinase 4 (PDK4). Of relevance, PDK4 expression has been shown to be enhanced in response to an acute bout of exercise, as well as in response to fasting in order to spare glucose when muscle glycogen levels are low [20]. Since keto diets are low in carbohydrates, they may result in a lack of glucose available to build muscle glycogen, which could activate PDK4. PDK4 expression may be also increased as a result of the high-fat nature of a keto diet, which can activate PPAR (peroxisome proliferator-activated receptor) α and induce the transcription of PDK4 [21]. It is unknown whether ketones (which are derived from fatty acids) can exert the same effect. In any case, several studies have reported that PDK4 expression was increased in the muscle in response to a keto diet or endurance exercise. Namely, in the gastrocnemius muscle collected from male mice following a four-week keto diet (with an unspecified composition) and both a single exhaustive treadmill running and weight-bearing swimming test, the mRNA expression of *Pdk4* was upregulated relative to a control diet [22]. *Pdk4* gene expression was also increased in the quadriceps muscle of male mice on a six-week high-fat keto diet (83.9% fat, 16.1% protein and 0% carbohydrates) relative to a control diet (although there was no additional effect of a three-week treadmill running regimen, nor was there any differences in PDH activity or pyruvate oxidation) [23].

PDK4 protein expression was increased with both diet and swimming exercise in male rats on the long-chain TAG keto diet described above (relative to the medium-chain TAG or control diet) [12]. However, no effect was observed with the medium-chain TAGs in the epitrochlearis muscle. As such, the authors proposed that a medium-chain TAG keto diet may avoid inhibiting the glycolytic pathway. Based on this proposition, it is possible that PDK4 is only activated in response to low glycogen levels, rather than a keto diet or exercise themselves. Since medium-chain TAGs are more rapidly oxidized by the liver, muscle glycogen could be somewhat spared due to quicker energy derivation from fatty acid oxidation and ketogenesis, which could explain why the diet did not alter PDK4 activity (relative to long-chain TAGs). The results showing increased PDK4 activity were maintained in a clinical study, which found that a 10-day keto diet (80% fat, 15% protein and 5% carbohydrates) increased PDK4 protein content in the vastus lateralis muscle of participants who perform at least 6 h of endurance exercise per week (relative to a control diet) [24]. However, interestingly, 10 days of exogenous D-βOHB supplementation did not increase PDK4 expression in a similar group of participants, which raises the possibility that the exogenous effect of ketones does not parallel the endogenous effects. It should be noted that the participants consumed a control (i.e., not keto) diet while taking the supplements, providing them access to carbohydrates. This supports the hypothesis outlined above that PDK4 is increased in response to depleted muscle glycogen and not a keto diet or supplementation. Consistent with this hypothesis is a study that concluded that incubating muscle with ketones does not allow for PDH activation. In isolated epitrochlearis muscle from male mice following a single bout of swimming, there was no impact of a 2 h incubation with 4 mM D,L-βOHB on the phosphorylation of PDH at Ser293 (an indicator of an activated state) [25]. A final study reported no change in PDK4 in response to ketones and exercise. Specifically, *Pdk4* mRNA expression was not altered in the mitochondria isolated from the gastrocnemius muscle of male mice on a six-week keto diet (69.5% fat, 20.2% protein, 10.3% carbohydrates) combined with a resistance running wheel exercise regimen [26]. This study differs from those presented above with respect to the type of exercise (resistance rather than endurance), which may indicate that endurance exercise is also linked to changes in PDK4 expression. It is also important to note that the diet in this study was higher in carbohydrates (10% vs. 0–5% in studies discussed above), which could theoretically result in a lower muscle glycogen depletion in response to exercise and thus, no effect in PDK4 expression.

Overall, studies that report no change in PDK4 expression involve either resistance exercise or ketone supplementation. Meanwhile, endurance exercise is consistently implicated in studies reporting increases in PDK4 expression in subjects or rodents on a keto diet. Therefore, parameters such as the type of exercise or the composition of the diet may influence PDK4 expression. In any case, it remains unclear whether increased PDK4 expression connected to a keto diet or exercise leads to an impact on PDH activity or glucose metabolism, a question that could be investigated in future studies. Further, whether decreased muscle glycogen is in fact the mechanism for increases in PDK4 expression in response to a keto diet and/or an exercise training regimen should be clarified.

## 4. Muscle Glycogen Stores Following Exercise and Exogenous or Endogenous Ketone Supplementation

Muscle glycogen is a primary energy source during high-intensity exercise and is typically depleted following an intense exercise bout or a very long endurance exercise [27]. Comparably, prolonged periods of decreased carbohydrate availability due to fasting are known to decrease muscle glycogen levels [28]. To begin, it has generally been concluded that endurance exercise training protects against the loss of muscle glycogen due to a keto diet. For example, after four weeks on a keto diet (88% fat, 11% protein and 1% carbohydrates), glycogen in the quadriceps of male mice was decreased relative to a control diet [11]. However, when the diets were extended by an additional eight weeks in combination with regular treadmill running (5 times per week, 30 min each time), there were no differences in muscle glycogen in mice on the keto diet and exercise program versus the control diet and exercise program (measured at least 24 h after the last bout of exercise). Therefore, it appears that the endurance program protected against muscle glycogen loss due to the keto diet. Consistently, in a comparable study using a six-week keto diet (83.9% fat, 16.1% protein and 0% carbohydrates) with a three-week treadmill running program during the second half of the diet (5 times per week, 1 h each time), training increased quadriceps glycogen levels in both diets but there was no difference between diets [23]. Similarly, in male endurance runners consuming a low carbohydrate diet for at least six months (>60% fat, <20% carbohydrates), there were no differences in muscle glycogen at rest, following a 180 min endurance running test (64% VO_2_max), nor following a subsequent 2 h recovery period relative to participants who had been consuming a high carbohydrate diet [29]. Overall, this group of studies suggests that increased ketolysis in combination with endurance training does not alter muscle glycogen levels relative to a control diet and the same training program. To the same point, in the muscle of the male rats given the keto diets with medium or long-chain TAGs described in previous sections, glycogen in the epitrochlearis muscle was lower with both keto diets relative to a control diet [12]. However, with swimming exercise, only the long-chain TAG diet decreased muscle glycogen relative to a control diet and exercise (measured at least 20 h following the last bout of exercise). This finding is consistent with the conclusions outlined above, since endurance exercise protects against muscle glycogen loss due to a medium-chain TAG diet. However, the fact that the same conclusion was not reported with the long-chain TAG diet supports the hypothesis outlined in the previous section that medium-chain TAGs spare muscle glycogen, and therefore, did not alter PDK4 expression.

On the other hand, a 12-week keto diet (with an unspecified composition) in combination with a 2-week training program (high-intensity interval training and periodized resistance and power exercises, plus endurance training once a week) resulted in decreased muscle glycogen in a group of military personnel, but stable levels in individuals who consumed a control diet with the exercise regimen [30]. The exercise program involved mainly resistance activities, which may explain why the conclusion differs from the effect of endurance exercise and a keto diet on muscle glycogen levels outlined above. Of note, in the previous section, it was postulated that PDK4 expression was increased with endurance exercise, and in response to low glycogen levels. This is contradictory to the general conclusion in this section that endurance exercise may spare muscle glycogen. Therefore, muscle glycogen levels and PDH should be investigated in the same studies or subjects going forward to better understand the effect of ketones on glucose metabolism.

For comparison, a second set of studies investigated the impact of endogenous ketone supplementation and exercise on muscle glycogen levels. For example, the addition of ketones to a carbohydrate-rich beverage consumed before and during a 2 h bicycle exercise at 45% VO_2_max, allowed male endurance athletes to have a smaller decrease in their muscle glycogen reserves [7]. Similarly, the consumption of a drink with D-βOHB allowed male athletes to replenish their muscle glycogen levels faster following an overnight fast and an interval cycling exercise [31]. Likewise, the regular consumption of a ketone ester drink in physically active males who followed a three-week endurance and high-intensity interval training allowed for maintenance of their muscle glycogen levels following a 30 min time trial while muscle glycogen was decreased in participants who did not receive the ketone ester drink [32]. In contrast to the studies with a keto diet detailed above, this group of studies suggests that exogenous ketone supplementation can spare muscle glycogen loss due to either one exercise bout or exercise training. With that being said, a pair of studies indicate that ketone ester supplementation had no effect on muscle glycogen. One study subjected physically active males to a glycogen-depleting exercise involving the leg, then provided them with a ketone ester drink and found that this did not alter muscle glycogen levels at the end of a 5 h recovery period [33]. The same conclusion was made at the end of a 3 h stimulated cycling race by male cyclists, who received D-βOHB before and during the activity [34]. To determine the direct effect of ketones on muscle glycogen levels, isolated epitrochlearis muscles from male mice following a single 60 min swimming exercise were exposed to 1, 2 and 4 mM of D,L-βOHB for 2 h. Only 4 mM of D,L-βOHB increased glycogen levels [25]. Therefore, it is possible that a higher concentration of ketones in the muscle is needed to influence muscle glycogen levels, whereas lower concentrations do not exert the same effect.

In conclusion, the effect of ketones and exercise on muscle glycogen levels are variable and appear to depend particularly on whether ketone body levels are being increased exogenously or endogenously. Additionally, it is possible that ketones have a signaling effect during exercise, possibly to increase the storage of glycogen during the recovery phase or decrease the use of muscle glycogen during the activity, therefore reducing the loss of glycogen. This hypothesis would also support the postulation outlined in the previous section that PDK4 is activated in response to low muscle glycogen levels.

## 5. Increased Ketolysis as a Potential Mechanism to Alter Exercise Performance

There have been several investigations into the effect of keto diets or exogenous ketone supplementation on exercise performance, with mixed results. To begin, studies implicating human participants and endurance exercise tests have reported a decrease in exercise performance following a keto diet. Namely, a three-day keto diet (50% fat, 45% protein and 5% carbohydrates) decreased mean power output during two 30 s tests on an exercise bike in male participants [35] and a 4-week keto diet (77% fat, 19% protein and 4% carbohydrates) decreased time to exhaustion on an incremental cycling test (an increase of 30 W every 4 min until 120 W) in female participants [36]. These results were maintained in physically active participants following a 10-day keto diet (80% fat, 15% protein and 5% carbohydrates): performance on an incremental cycling test (90 min at 70% VO_2_max, followed by incremental increases to fatigue) was decreased relative to a control diet [24]. On the other hand, an eight-week keto diet (approximately 68% fat, 25% protein and 5% carbohydrates) did not change performance on bench press and squat tests in male bodybuilders [37]. Since this study measured the effect of the keto diet on resistance exercise performance, this could explain why the conclusion differs from the endurance-type studies outlined above. Next, in human participants, an eight-week cyclical keto diet (keto diet on weekdays, high carbohydrate diet on weekends) combined with regular strength and aerobic workouts increased performance on certain strength exercises (relative to baseline) [38]. The improvement associated with the cyclical keto diet may be due to the workout program administered to the participants rather than the diet itself, but the cyclical design could warrant further investigation in comparison to a continuous keto diet with respect to exercise performance.

Secondly, studies using rodents have shown that a period of increased ketolysis had no impact on endurance exercise performance. As an example, a four-week keto diet (with an unspecified composition) in male mice did not change running distance or time on an exhaustive treadmill running test, nor swimming time on an endurance swimming test [22]. Similarly, while male rats consumed a six-week keto diet (69.5% fat, 20.2% protein and 10.3% carbohydrates) and had access to a voluntary running wheel, there was no difference in cumulative running distance relative to mice on a control diet [39]. The same conclusion was also drawn in female mice following an 8 h fast to induce ketosis: exercise duration in an endurance treadmill running test was unchanged relative to mice who had not fasted [40]. To finish summarizing the reported effect of a keto diet on exercise performance, one report was made that a keto diet may contribute to improved exercise performance. Specifically, an eight-week keto diet (76.1% fat, 8.9% protein and 3.5% carbohydrates) led to a longer running time in male rats during an exhaustive running test (relative to a control diet) [41]. The keto diet lasted longer than the regimens described above, which may indicate that longer-term keto diets are needed to influence endurance capacity in rodents.

For comparison, a selection of studies also measured the effect of endogenous ketone supplementation on exercise performance. In the study cited above that reported a decrease in exercise capacity in active participants on a 10-day keto diet, supplementation of a regular diet with D-βOHB for 10 days had no effect on performance in the cycling test [24]. Similarly, consumption of D-βOHB by male cyclists did not affect power output during a subsequent 3 h stimulated race [34], a conclusion that was replicated in another group of male cyclists administered a ketone ester drink during the race [42]. Together, these studies suggest that ketone supplementation does not affect endurance performance in human participants. Contrarily, in physically active males, three weeks of endurance and high-intensity interval training in combination with a ketone ester drink resulted in increased power output in the final 30 min of a 2 h endurance activity [32]. Likewise, two weeks of D,L-βOHB supplementation in male mice resulted in increased distance, time to exhaustion and maximal speed on a weekly treadmill test [43]. Yet, this increase was not maintained after a total of six weeks. Two hypotheses can be drawn from this conclusion: the effect of supplementation with D-βOHB differs from that of D,L-βOHB and that ketone supplementation may only have a short-term influence on exercise performance. An investigation using male endurance athletes also found that consuming a beverage with 40% of calories coming from ketone esters (relative to 100% carbohydrates) led to increased distance covered in a time trial that was preceded by a 60 min cycle [7]. Although the increase was minor (2%), the increased carbohydrate composition of the beverage may be what allowed for the positive impact on endurance exercise performance.

Interestingly, one study took a thoughtful approach and measured exercise capacity in male mice with a muscle-specific SCOT knockout. In this case, the knockout mice ran for the same time and distance during a time-to-exhaustion treadmill test as wild-type mice, implying that preventing muscle ketone body oxidation did not impact exercise capacity [44]. Ketones have also been found to exert signaling in addition to metabolic effects [45]; therefore, using muscle SCOT knockout mice would be an interesting addition to future investigations on the relationship between ketones and exercise, as it could assist in concluding whether ketone body oxidation is necessary for potential metabolic effects induced by exercise.

## 6. Alterations in AMPK and PCG1α Activation Following an Exercise and Keto Regimen

It is well known that AMP-activated protein kinase (AMPK) is an energy-sensing enzyme, that is activated by phosphorylation in most tissues, including the muscle, to generally stimulate energy-replenishing processes. Exercise is a prominent activator of AMPK, as are high levels of AMP relative to ATP during fasting [46]. Yet of relevance to this review, whether increased muscle ketolysis induces alterations in AMPK activation or activity has not been elucidated. However, some studies have investigated the relationship between AMPK, ketones and exercise in the muscle. To begin, one study found that a six-week high-fat, no-carbohydrate diet (83.9% fat, 16.1% protein and 0% carbohydrates) combined with a three-week treadmill running exercise regimen increased levels of phosphorylated AMPK in the quadriceps muscle of male mice relative to mice subjected to the same exercise regimen, but with a control diet [23]. In contrast, a single resistance exercise test following a six-week keto diet (69.5% fat, 20.2% protein and 10.3% carbohydrates) in male rats did not alter the levels of phosphorylated AMPK in the gastrocnemius muscle [39]. The variable length (three weeks versus a single bout) and type (endurance versus resistance) of the exercise, but also the difference in the diet composition (more carbohydrates in the second study) could explain the different conclusions between these studies in rodents. Furthermore, a clinical investigation of male volunteers involved in regular physical activity provided mixed results. The participants consumed a ketone ester drink during the recovery period following a glycogen-depleting exercise protocol in the leg, then muscle biopsies were taken to measure AMPK phosphorylation. AMPK phosphorylation was decreased 90 min post-exercise, but there was no difference 5 h post-exercise relative to participants who did not consume the ketone ester drink [33]. This result may be different than the increased phospho-AMPK levels observed in the rodent studies described above since the study used human participants, and they were administered a single ketone supplement following the bout of exercise (rather than a longer term keto diet which preceded muscle collection). Comparably, the epitrochlearis muscle was harvested from male mice following a 60 min swimming exercise, then subjected to 4 mM of D,L-βOHB ex vivo. Exposure to D,L-βOHB for 15 min decreased the phosphorylation of AMPK and ACC (acetyl-CoA carboxylase, a downstream target of AMPK), but not after 2 h [25]. As such, it is possible that increased exposure to ketones following exercise results in only a brief decrease in muscle AMPK activity, but as the ketone supply dwindles, AMPK activation begins to increase.

Comparably to AMPK, peroxisome proliferator-activated receptor-γ coactivator 1-alpha (PGC1α) is a transcriptional coactivator that regulates energy metabolism, and its expression is enhanced by exercise in the muscle [47]. Although the results are limited, the effect of a keto diet and exercise on muscle PGC1α has been investigated, with consistent results. *Pgc1α* gene expression was enhanced in the gastrocnemius and soleus muscle of male mice following an eight-week exercise program, but there was no additional effect of a keto diet (76.1% fat, 8.9% protein and 3.5% carbohydrates) combined with the exercise [8]. The same conclusion was drawn by a similar study that also used the gastrocnemius muscle of male mice following six weeks on a keto diet (69.5% fat, 20.2% protein and 10.3% carbohydrates) and an exercise program [26]. To a different conclusion, a six-week high-fat diet with no carbohydrates (83.9% fat, 16.1% protein and 0% carbohydrates) combined with three weeks of exercise increased *Pgc1α* gene expression in the quadriceps muscle of male mice, but the same effect was not seen with a control diet [23]. Since all three of these studies used male mice and a similar type of exercise (running on a treadmill or a resistance-loaded wheel), the variability may be due to the fact that the study which reported an increase in *Pgc1α* expression as a result of the keto diet did not subject the mice to the exercise program for the whole length of the keto diet (as was the case for the pair of studies reporting no effect of a keto diet on *Pgc1α*). In spite of these results, in the absence of studies investigating the effect of ketones themselves on the muscle expression of PGC1α (e.g., exogenous supplements or ex vivo exposure), it is unclear whether PGC1α is increased due to the high-fat component of a keto diet, low muscle glycogen levels or as a direct result of increased ketolysis or ketone levels.

Overall, it is clear that ketones and exercise can act on regulators of energy-sensing pathways (AMPK and PGC1α). However, it is not clear whether there is an additional effect when they are considered in combination, and factors such as the type and length of exercise, whether ketones are increased pre- or post-exercise, and the length of time post ketone supplementation should be taken into consideration in future studies.

## 7. Influence of Increased Exposure to Ketones on Glucose Metabolism

In a post-prandial period (when blood glucose is high), insulin signals through insulin receptors in the muscle, which leads to the phosphorylation and activation of a group of proteins including IRS1 (insulin receptor substrate) and AS160 (Akt substrate of 160 kDa) [9]. This enables translocation of GLUT4 (glucose transporter type 4) to the membrane, and glucose uptake into muscle cells. Given the shift away from glucose metabolism that is essential to a keto diet, it is important to consider the impact on enzymes connected to glucose oxidation. One study found that a 10-day keto diet (80% fat, 15% protein and 5% carbohydrates) in combination with at least 6 h of physical activity per week resulted in increased plasma glucose and insulin levels following a glucose tolerance test. Although not specific to the muscle, it raises the possibility that keto diets may have a negative impact on insulin sensitivity. In contrast, exogenous supplementation of a normal diet with ketones did not produce the same effect [24]. It is thus possible that the negative effect of the keto diet on glycemia was due to the increased lipid content of the diet, rather than the ketones themselves, and that the effect of exogenous ketones differs from that of a keto diet.

To begin, a collection of studies investigated the effect of keto diets on factors linked to glucose metabolism. Namely, in trained male cyclists who had been on low carbohydrate, high fat diets (with variable compositions) for at least six months: relative to participants on a control diet, GLUT4 and IRS1 protein content was decreased in the muscle [48]. On the other hand, another study found that ketones and exercise had no effect on muscle glucose metabolism. In particular, a 6-week keto diet in male mice found that regular running exercise increased GLUT4 gene expression in the gastrocnemius muscle, but that there was no additional effect of the keto diet relative to a control diet [26]. The shorter time frame for the keto diet could explain the difference in results: it is possible that keto diets are more detrimental to the muscle in the long term. Interestingly, exercise may protect against keto diet-related decreases in GLUT4 expression, as a 12-week keto diet (88% fat, 11% protein and 1% carbohydrates) without exercise resulted in decreased gene expression of GLUT4 and pyruvate kinase (which catalyzes the last step in glycolysis) in the quadriceps muscle of male mice, but this effect was counteracted by combining the diet with eight weeks of treadmill running [11]. Taken together, these studies indicate that keto diets may be negatively impacting factors related to muscle glucose metabolism. However, the persistence of this observation in the long term or following the cessation of the keto diet is not known, nor is whether exercise can counteract these potential negative effects. Further, it is well understood that high-fat diets can induce muscle insulin resistance, which is a cause for concern due to the fact that keto diets are high in fat, and therefore, a potential mechanism for any negative effects of a keto diet on glucose metabolism [49]. Another potential mechanism for these observations is that the low carbohydrate content of keto diets can influence GLUT4 protein levels or other parameters linked to glucose metabolism. In support of this postulation, previous studies have shown that while a high-fat diet decreased muscle GLUT4 expression, a high-calorie carbohydrate diet did not exert the same effect in mice [50]. Similarly, a four-week high carbohydrate, fat-restricted diet in combination with daily swimming exercise led to decreased PDK4 expression and increased muscle glycogen utilization in male rats during exercise relative to a control diet [51].

Next, the effect of ketone bodies themselves may differ from the effect of keto diets on muscle glucose metabolism. Namely, it was previously shown that administering a ketone ester to mice with obesity leads to improved glucose tolerance [52]. Although not connected to exercise, these results are interesting because the same conclusion was drawn in mice with obesity and a muscle-specific SCOT knockout. This suggests that ketone esters could exert a beneficial signaling effect on glucose metabolism, which contrasts with the potential negative effect of keto diets described above. These results were also maintained in individuals without obesity, where the administration of a ketone supplement prior to a meal decreased post-prandial glucose levels, which further highlights the potential signaling effect of exogenous ketones [53]. At the same time, ketone intake did not alter plasma insulin levels or gastric emptying, which suggests that the ketones positively impacted peripheral glucose uptake. Whether ketones improved peripheral insulin action or insulin-independent glucose uptake in peripheral tissues requires further investigation. However, other studies have reported that when exercise is introduced into the study design, ketone bodies themselves can have a negative effect on glycolysis and insulin signaling in the muscle. A study that provided male endurance athletes with a beverage supplemented with an increased number of calories from ketone esters during a bicycle test found that intramuscular glucose levels were increased at the end of the exercise (relative to when the participants consumed a beverage higher in carbohydrates) [7]. The authors propose that a state of ketosis leads to decreased muscle glycolysis, which was supported by decreased levels of fructose-1,6-bisphosphate and 1,3-bisphosphoglycerate (glycolytic intermediates) in the muscle as a result of the ketone-supplemented beverage (relative to the carbohydrate beverage). Comparably, in the epitrochlearis muscle of male mice following a 60 min swimming exercise, ex vivo exposure to 4 mM of D,L-βOHB resulted in increased AS160 phosphorylation after 15 min, but decreased AS160 phosphorylation after 2 h [25], suggesting that short exposure to D,L-βOHB increases GLUT4 translocation but longer exposure decreases it. Further investigations are warranted to determine how a potential positive effect of exogenous ketones can be exploited. Overall, before providing recommendations regarding keto diets or supplements for athletes, their impact on glucose metabolism should be investigated further.

## 8. Secretion of IL-6 Following a Keto Diet and Exercise Regimen

The muscle can act as an endocrine organ, by producing and releasing small peptides termed myokines. Myokines can be spontaneously released while at rest, but exercise (whether acute or chronic) regulates myokine secretion [54]. As an example, interleukin (IL)-6 is a myokine known to be dramatically increased in the circulation in response to exercise and has also been linked to glucose metabolism in the muscle during exercise [55]. Specifically, it was found that IL-6 was increased in response to low glycogen levels in human muscle [56], which led to a postulation that it acts as an energy sensor during exercise (and by extension, this potential effect may be applicable to a general state of carbohydrate deprivation) [55]. With this in mind, a pair of studies suggest that ketones and exercise may alter IL-6 levels. In male bodybuilders, an eight-week keto diet (approximately 68% fat, 25% protein and 5% carbohydrates) decreased plasma IL-6, in contrast to a control diet which increased plasma IL-6. However, plasma levels do not necessarily reflect the quantity of IL-6 secreted from the muscle since other tissues also secrete this cytokine [37]. A second study measured IL-6 levels specifically in the soleus muscle of male mice administered a keto diet for eight weeks (76.1% fat, 8.9% protein and 3.5% carbohydrates), with or without regular exercise, and found that IL-6 mRNA was significantly increased as a result of exercise, an effect that was enhanced with the keto diet [8]. Interestingly, this effect was not maintained in the gastrocnemius muscle of the same mice, suggesting that the effect of the keto diet on IL-6 expression may be specific to oxidative muscles. Also, the plasma levels of IL-6 did not parallel the levels in the muscle: although plasma IL-6 was also increased with exercise, a control diet enhanced this effect. Overall, the potential connection of IL-6 to a keto diet in combination with exercise warrants further clarification. Similarly, alterations in the secretion of other contraction-induced myokines in response to a keto diet should also be investigated. When doing so, it should be taken into consideration that the high-fat requirement of keto diets may induce inflammation, which could be the mechanism for increased production of IL-6 or other myokines tied to an inflammatory response.

## 9. Muscle TAG Levels Following Exercise and a Keto Diet or Ketone Supplementation

The muscle has lipid droplets which store TAGs, which can be metabolized to generate fatty acids and ultimately ATP through fatty acid oxidation. These muscle TAG stores are typically reduced following exercise [57], but very few studies have investigated the impact of ketones and exercise on muscle TAGs, and those that have present conflicting results. In a group of male cyclists and triathletes who received D-βOHB prior to a stimulated cycling race, intramuscular TAGs were unchanged in a muscle biopsy following the test [34]. However, in male endurance athletes, consuming a beverage with more calories from ketone esters resulted in a greater decrease in intramuscular lipids following a fixed-intensity bicycle exercise (relative to consuming a drink higher in carbohydrates) [7]. Lastly, in military personnel on a 12-week keto diet (with an unspecified composition) combined with two weeks of exercise training, intramuscular TAGs increased at the end of the program, whereas the levels decreased in individuals on a control diet [30]. The type of exercise in this study differs from the others outlined in this section, as the participants were mainly subjected to regular high-intensity interval training and periodized resistance and power exercises, rather than an endurance cycling test. Similarly, the participants followed a keto diet rather than taking ketone supplements. It is thus possible that the higher muscle TAG content with the keto diet was the result of an increased consumption of fat rather than increased muscle ketolysis or a signaling effect of the ketones themselves. Going forward, the possible effect of a keto diet on muscle TAG levels should be taken into consideration, as a surplus of muscle TAGs has been linked to conditions such as insulin resistance when the individuals are inactive [58,59].

## 10. Impact of Ketones on Muscle Health Following Exercise

Finally, a variety of studies have investigated the impact of ketones in combination with exercise on parameters related to overall muscle health in response to exercise. To begin with recovery following exercise, a study in male mice found that an eight-week keto diet (76.1% fat, 8.9% protein and 3.5% carbohydrates) led to an accelerated recovery phase following a treadmill endurance test (measured by the amount of movement 24 h following the test) [60]. However, another study had a group of recreational athletes perform a series of eccentric knee extensors following an overnight fast. The participants drank a D-βOHB beverage over the course of the day of the activity and for the two days following it. Muscle soreness (evaluated subjectively) and muscle function (evaluated by repeating the test) were not affected by the ketone ester supplementation during the recovery period [61]. Similarly, a six-week keto diet (69.5% fat, 20.2% protein and 10.3% carbohydrates) in male rats leading up to a bout of running on a resistance-loaded running wheel had no effect on post-exercise muscle protein synthesis [39]. Although there are only three studies investigating the potential impact of ketones on muscle recovery and they had variable designs and methods for quantifying recovery, it is possible that the type of exercise has an effect, since no impact on muscle recovery was reported with activities with an increased eccentric demand (i.e., the knee extensors and resistance running described above).

Some investigations were also made into the effect of a keto diet and exercise on markers of muscle damage, particularly plasma levels of lactate dehydrogenase (LDH) and creatine kinase (CK). For example, an eight-week keto diet (76.1% fat, 8.9% protein and 3.5% carbohydrates) did not protect against increased plasma levels of LDH and CK 72 h following an endurance running test in male mice [41], nor did the consumption of tablets containing βOHB prior to a 30 min bout of downhill running in male volunteers [62]. Contrarily, in male cyclists who consumed a keto diet (55% fat, 35% protein and 10% carbohydrates) for approximately one week, then a ketone-supplemented tablet prior to a cycling session, plasma CK and LDH were not increased 120 min post-exercise, as was observed in the control group [63]. It is possible that the difference in the plasma collection time post-exercise explains the variable results between these studies. Additionally, the type of exercise may play a role in the effect of a keto diet on muscle damage, as for example, downhill running has been shown to induce more muscular damage than flat and uphill running due to increased eccentric contractions [64].

Next, muscle weight following a keto diet was also measured in one study but should be investigated in more detail in future experiments given the conclusion. Specifically, in participants with obesity, an eight-week low carbohydrate diet (<50 g per day) with or without exercise decreased lean muscle mass relative to a control diet [65]. Lastly, the consumption of a ketone supplement beverage had no effect on muscle tissue oxygenation in male distance runners following a voluntary hypoventilation protocol [66], and only slightly (3%) increased muscle oxygenation in a group of male cyclists who consumed a ketone ester drink during a 3 h race [42]. However, administration of a ketone ester drink for three weeks, plus regular endurance training in physically active males did improve several parameters related to muscle angiogenesis [67]. The same protocol had previously been shown to induce a positive effect on exercise performance, suggesting that increased muscle angiogenesis may be a mechanism for improved exercise performance due to ketone supplements [32].

Overall, the effect of ketones and exercise on muscle recovery, damage, weight and oxygenation have not been studied in enough depth to draw conclusions, but they are parameters that could be taken into consideration going forward.

## 11. Summary

As outlined above, there is a multitude of variables to consider when analyzing the effect of ketones and exercise on the muscle. This includes but is not limited to the muscle type being investigated (e.g., oxidative versus glycolytic), the type of exercise (e.g., acute versus chronic, aerobic versus resistance), the length and composition of the keto diet (e.g., medium versus long-chain TAGs, D versus L-βOHB) or the comparative effect of exogenous ketone supplementation. Given the lack of research conducted in this area and the variability between studies with respect to these parameters, it is often difficult to draw clear conclusions based on the current literature regarding the relationship between ketones and exercise in the muscle. Yet, there is evidence to suggest that exercise increases parameters related to ketone body metabolism (e.g., ketolytic gene expression), that PDK4 expression increases in response to low muscle glycogen, rather than as a direct consequence of a keto diet or exercise, and that endurance exercise may spare muscle glycogen loss due to increased exposure to ketones. It should be noted, however, that these are only postulations based on the work conducted to date, so future experiments are needed in order to validate these hypotheses. The effect of ketones and exercise on other factors, such as exercise performance, myokine secretion and muscle TAG levels, remain more ambiguous and should be taken into consideration when designing experiments in the future. The effect of exercise in combination with either ketone supplements or a keto diet on all the parameters linked to the muscle discussed in this review is summarized in Figure 1 and Table 1.

## 12. Limitations and Future Direction

A major limitation of this review is the overall lack of research studies that investigate the connection between muscle, exercise and ketones. Many studies investigate two of these factors, but few cover all three. This issue is further compounded by the high variability in the design of studies pertinent to the goal of this review. Additional limitations and recommendations for future studies are outlined below.

Firstly, it should be noted that the high-fat requirement of a keto diet may lead to negative consequences in the muscle that are not due to increased ketone exposure in itself (e.g., inflammatory responses). Future experiments which further investigate the mechanisms for effects connected to a keto diet would assist in making this distinction. To this point, continuing to explore the difference between keto diets and exogenous ketone supplementation may be advantageous, because supplements eliminate the need for high levels of dietary fat. As mentioned in the section on the effect of keto diets on glucose metabolism, it is also possible that the low carbohydrate requirement of a keto diet is negatively impacting GLUT4. Future studies could employ a high carbohydrate diet as a control group to better elucidate the impact of carbohydrate deprivation on the muscle.

Going forward, separate study groups should be implemented to clearly investigate the effect of a keto diet or exogenous ketone supplementation alone, exercise alone and a combination of both variables. This will allow for clearer determinations into their effect on the muscle, as many studies published to date do not make these distinctions (for example, they only use participants who are physically active). Hence, clinical studies should also make use of individuals who are sedentary, as the effect of ketones may be variable in individuals without a significant background in physical activity. Increasing the diversity of study designs and the associated participant profiles is necessary to obtain a more comprehensive understanding of the relationship between ketone bodies and exercise in the muscle.

Similarly, female participants or animals should be used more frequently, and contrasted to male subject groups, as almost all the research presented here used exclusively male subjects/rodents. It is crucial that research in this area becomes more inclusive, as the effect of increased ketolysis may be variable between biological sexes, and recommendations regarding keto diets and supplements may be vastly different for males and females.

In general, before recommendations are made regarding keto diets or exogenous ketone supplementation, the long-term impact of these regimens on muscle metabolism, particularly glucose oxidation, should be determined.

## 13. Conclusions

Overall, as summarized above, there is evidence to suggest that increased exposure to ketones can influence (whether positively or negatively) muscle metabolism, but a significant number of important questions pertaining to this area of research remain outstanding. As the popularity of keto diets and ketone supplements continues to rise, an increasing number of individuals are augmenting their exposure to ketone bodies through their diet, athletes may be implementing the regimens into their training programs and doctors are recommending these interventions to their patients. However, until more work is conducted on the relationship between ketones, the muscle and exercise, it is uncertain whether these recommendations and lifestyle alterations are beneficial or detrimental. As such, future investigations into the connection between exercise and keto diets or exogenous ketone supplementation is strongly encouraged, especially as they relate to skeletal muscle metabolism.

## Figures and Tables

**Figure 1 nutrients-15-04228-f001:**
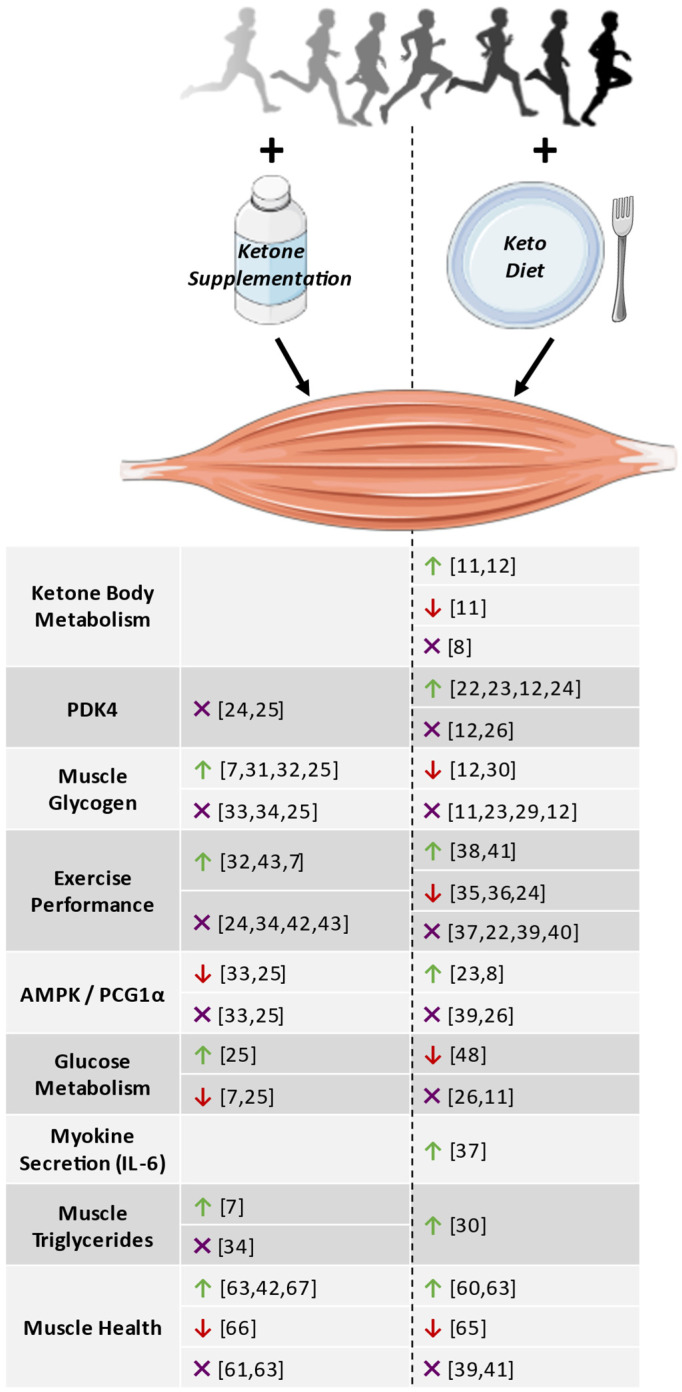
Summary of articles investigating the effect of exercise with either ketone supplements or a keto diet on various parameters in the muscle. Green ↑ = reported an increase or positive effect relative to control groups, red ↓ = decrease or negative effect, purple X = no effect. The figure was generated in part by using Servier Medical Art, which is provided by Servier and licensed under a Creative Commons Attribution 3.0 unported license, as well as rawpixel, which is licensed under a Creative Commons 1.0 Universal Public Domain Dedication.

**Table 1 nutrients-15-04228-t001:** Summary of the current literature on the effect of ketone bodies and exercise on muscle metabolism.

Reference	Study Population	Study Design	Key Finding(s)
[7]	Male endurance athletes	■ Consumption of ketone ester beverage during exercise test (2 h bicycle test at 45% VO_2_max)	■ Increased muscle D-βOHB levels pre-exercise and 1 h post-exercise ■ Reduced exercise-induced decrease in muscle glycogen reserve ■ Increased distance covered in a time trial after 60 min of cycling ■ Increased intramuscular glucose levels post-exercise ■ Greater decrease in intramuscular lipids due to exercise
[8]	Male mice	■ 8-week keto diet (76.1% fat, 8.9% protein, 3.5% carbohydrates) ■ 8-week exercise regimen	■ No effect on soleus or gastrocnemius *Oxct1* expression ■ Keto diet alone increased *Bdh* expression in soleus and decreased *Bdh* expression in gastrocnemius ■ Exercise increased gastrocnemius and soleus *Pgc1α* expression ■ Combination of diet and exercise increased soleus but not gastrocnemius IL-6 mRNA
[11]	Male mice	■ 12-week keto diet (88% fat, 11% protein, 1% carbohydrates) ■ 8 weeks treadmill exercise	■ Keto diet alone decreased gastrocnemius *Oxct1* expression, but no additional effect of exercise ■ Combination of diet and exercise increased gastrocnemius *Hmgcs2* expression ■ Keto diet decreased muscle glycogen and GLUT4 and pyruvate kinase gene expression, which were all reversed with the exercise regimen
[12]	Male rats	■ 8-week keto diet (comparison of medium and long-chain TAGs) ■ 8-week swimming exercise	■ Exercise increased epitrochlearis SCOT expression ■ SCOT expression increased with medium chain relative to long-chain TAGs ■ Increased PDK4 expression with long but not medium-chain TAGs ■ Both diets decreased muscle glycogen, which was reversed with exercise only for the medium-chain diet
[13]	Male and female mice	■ Single bout of treadmill running	■ No effect on quadriceps *Bdh*, *Oxct1* or *Acat* expression ■ Increased plasma ketones in female versus male mice
[14]	Male rats	■ 10-week swimming exercise	■ Increased gastrocnemius SCOT activity
[15]	Male rats	■ 12-week treadmill running	■ Increased gastrocnemius SCOT, BDH and ACAT activity; and ■ Increased uptake of acetoacetate and D,L-βOHB in gastrocnemius
[16]	Male rats	■ 15-week treadmill running	■ Decreased SCOT activity in diaphragm ■ No effect on SCOT activity in intercostal muscle
[17]	Male rats	■ 14–28 weeks treadmill running	■ Increased uptake of acetoacetate and βOHB together in perfused hindlimb muscle ■ No effect on uptake of βOHB alone
[18]	Male rats	■ Single bout of treadmill running	■ No effect on acetoacetate or βOHB-supported respiration in permeabilized gastrocnemius
[19]	C2C12 cells	■ Exposure to forskolin to mimic exercise	■ Increased βOHB in cell lysates and culture media
[22]	Male mice	■ 4-week keto diet (unspecified composition) ■ Single exhaustive treadmill and weight bearing swimming test	■ Increased gastrocnemius *Pdk4* expression ■ No change in running distance or time, or in swimming time
[23]	Male mice	■ 6-week high fat keto diet (83.9% fat, 16.1% protein, 0% carbohydrates) ■ 3 weeks treadmill running	■ Increased quadriceps *Pdk4* expression due to keto diet ■ No additional effect of exercise and no effect on PDH activity or pyruvate oxidation ■ Training increased glycogen levels independently of exercise ■ Increased phosphorylation of AMPK in the quadriceps ■ Increased *Pgc1α* expression
[24]	Participants performing >6 weekly hours of endurance exercise	■ 10-day keto diet (80% fat, 15% protein and 5% carbohydrates) or D-βOHB supplementation	■ Keto diet increased vastus lateralis PDK4 expression ■ No effect of supplements on PDK4 ■ Keto diet decreased performance on incremental cycling test (90 min at 70% VO_2_max, followed by incremental increases to fatigue), but supplements had no effect ■ Keto diet, but not ketone supplements, increased plasma insulin and glucose levels following glucose tolerance test
[25]	Epitrochlearis from male mice	■ Muscle harvested after swimming for 60 min ■ 2 h incubation with 4 mM D,L-βOHB	■ No impact on PDH phosphorylation at Ser293 ■ Increased glycogen levels ■ Decreased AMPK and ACC phosphorylation after 15 min, but no difference after 2 h ■ AS160 phosphorylation increased after 15 min, but decreased after 2 h
[26]	Gastrocnemius from male mice	■ 6-week keto diet (69.5% fat, 20.2% protein and 10.3% carbohydrates) ■ 6 weeks on resistance running wheel	■ No effect on *Pdk4* expression ■ Increased *Pgc1α* expression due to exercise ■ Exercise increased muscle *Glut4* gene expression
[29]	Male endurance runners	■ Low carbohydrate diet (>60% fat, <20% carbohydrates) for >6 months	■ No difference in muscle glycogen following an endurance running test and 2 h recovery period
[30]	Military personnel	■ 12-week keto (unspecified composition) ■ 2-week mixed training program	■ Decreased muscle glycogen ■ Increased intramuscular TAGs
[31]	Male athletes	■ Overnight fast and interval cycling exercise ■ Consumption of D-βOHB beverage	■ Muscle glycogen was replenished more rapidly
[32]	Physically active males	■ 3-week endurance and interval training ■ Regular consumption of ketone ester drink	■ Maintenance of muscle glycogen following 30 min time trial ■ Increased power output in the final 30 min of a 2 h endurance activity
[33]	Physically active males	■ Glycogen depleting exercise in the leg, then consumption of ketone ester drink	■ No change in muscle glycogen after 5 h recovery period ■ Decreased muscle AMPK phosphorylation 90 min post-exercise, but no difference after 5 h
[34]	Male cyclists	■ Consumption of D-βOHB before and during a 3 h race	■ No change in muscle glycogen after the exercise ■ No effect in power output during the race ■ No effect on intramuscular TAGs
[35]	Male participants	■ 3-day keto diet (50% fat, 45% protein and 5% carbohydrates)	■ Keto diet decreased mean power output in two 30 s exercise bike tests
[36]	Female participants	■ 4-week keto diet (77% fat, 19% protein and 4% carbohydrates)	■ Keto diet decreased time to exhaustion on ingle incremental cycling test (increase of 30 W every 4 min until 120 W)
[37]	Male bodybuilders	■ 8-week keto diet (approximately 68% fat, 25% protein and 5% carbohydrates)	■ No effect on performance on bench press and squat tests ■ Decreased plasma IL-6
[38]	Male and female participants	■ 8-week cyclical keto diet (keto diet on weekdays, high carbohydrate diet on weekends) with regular strength and aerobic workouts	■ Increased performance on certain strength exercises
[39]	Male rats	■ 6-week keto diet (69.5% fat, 20.2% protein and 10.3% carbohydrates)	■ No effect on running distance on a voluntary running wheel throughout the 6 weeks ■ No effect on phosphorylation of AMPK in gastrocnemius ■ No effect on post-exercise muscle protein synthesis
[40]	Female mice	■ 8 h fast to induce ketosis	■ No effect on endurance in treadmill running test
[41]	Male rats	■ 8-week keto diet (76.1% fat, 8.9% protein and 3.5% carbohydrates)	■ Longer running time in male rats during an exhaustive running test ■ Did not protect against increased plasma levels of LDH and CK 72 h post endurance running
[42]	Male cyclists	■ Administration of ketone ester drink during stimulated cycling race	■ No impact on power output during the race ■ Increased muscle oxygenation during the race
[43]	Male mice	■ 6 weeks of D,L-βOHB supplementation	■ Increased distance, time to exhaustion and maximal speed on a weekly treadmill test only after 2 weeks
[44]	Male SCOT knockout mice	■ Time to exhaustion treadmill test	■ No difference in running time and distance
[48]	Male cyclists	■ Low carbohydrate, high fat diet (variable compositions) for at least 6 months	■ Decreased muscle GLUT4 and IRS1
[51]	Male rats	■ 4-week high carbohydrate, fat-restricted diet and daily swimming exercise	■ Decreased PDK4 expression and increased muscle glycogen utilization
[60]	Male mice	■ 8-week keto diet (76.1% fat, 8.9% protein and 3.5% carbohydrates)	■ Accelerated recovery phase following a treadmill endurance test
[61]	Recreational athletes	■ Participants performed eccentric knee extensors following overnight fast ■ Also consumed a D-βOHB beverage on the day of the activity, and for the 2 days after	■ Did not affect muscle soreness and muscle function during the recovery period
[62]	Male participants	■ Consumption of βOHB tablets prior to 30 min of downhill running	■ No effect on plasma LDH and CK post-exercise
[63]	Male cyclists	■ Keto diet (55% fat, 35% protein and 10% carbohydrates) for 1 week, and ketone supplemented tablet prior to cycling session	■ No effect on plasma LDH or CK 120 min post-exercise
[65]	Participants with obesity	■ 8-week low carbohydrate diet (<50 g per day) with regular exercise	■ Keto diet decreased lean muscle mass
[66]	Male distance runners	■ Consumption of ketone supplemented beverage ■ Voluntary hypoventilation protocol	■ No effect on muscle tissue oxygenation
[67]	Physically active males	■ Ketone ester drink and regular endurance training for 3 weeks	■ Improved parameters related to muscle angiogenesis

## Data Availability

No new data were created in this review article.

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
