# Peer review of "Exogenous Ketone Supplementation and Ketogenic Diets for Exercise: Considering the Effect on Skeletal Muscle Metabolism"

_nutrients, 2023, doi:10.3390/nu15194228_

Round 1
Reviewer 1 Report
Major comments
Overall points.
In this review, the authors discussed the effects of ketogenic diets and exogenous ketone supplementation on skeletal muscle metabolism and exercise. The authors reviewed the literature widely, then I learned a lot from this manuscript.
However, in some parts discussing influences of ketogenic diet consumption, whether those effects were due to ketone body per se or other factors induced by high fat exposure and/or low carbohydrate level were not sufficiently organized in the manuscript. I think this manuscript would become better if this point was improved.
2. Effects of exercise on muscle ketone body metabolism
All studies referred in this section were performed in animals or cells and it seems that no investigation related to the effects of ketone bodies on enzymatic activities has been performed in human subjects. I recommend mentioning it as a limitation.
3. Muscle PDK4 activation following a keto diet and exercise.
Please provide background information on why you focus on the regulation of pyruvate dehydrogenase regulation in this review. How did you hypothesize that ketone bodies influence PDH regulation?
6. Alterations in AMPK and PCG1α activation following an exercise and keto regimen
According to the second paragraph, it seems that the upregulation of PGC1a expression in skeletal muscle induced by a ketogenic diet was due to high fat exposure and low glycogen level. Is there any study that examined the effects of ketone bodies per se on PGC-1a expression in skeletal muscle (or C2C12 cells)?
7. Influence of increased exposure to ketones on glucose metabolism
A ketogenic diet is a very low carbohydrate diet. It has been suggested that the percentage of carbohydrates in diet would influence skeletal muscle GLUT4 expression (for example, PMID: 8419118, 11473054, 33456007). Moreover, it is well known that high fat (low carbohydrate) diet induces insulin resistance. I recommend that the authors reconstitute this section to organize the effects of ketone bodies per se and a ketogenic diet (= very low carbohydrate diet) on glucose metabolism in skeletal muscle.
Many recent studies have shown that ketone monoester intake attenuated the increase in blood glucose levels in OGTT tests (e.g. PMID: 33010116, 31599919, 32941737). Some of these reported that ketone monoester intake did not influence plasma insulin level or gastric emptying, suggesting that ketone body would impact on peripheral insulin action. If appropriate, I recommend discussing these studies in this section.
Lines 456-460 Please show the references of these two sentences.
10. Impact of ketones on muscle health following exercise
In this section, I recommend reviewing reference 32 and the subsequent study of that (PMID: 37062892). Reference 32 investigated effects of ketone monoester intake after the daily training and they showed the group who took ketone monoester after exercise could train with a greater work output. The same group showed in PMID: 37062892 that this might stimulate exercise-induced adaptation in skeletal muscle.
Author Response
Response to R1
Overall points.
In this review, the authors discussed the effects of ketogenic diets and exogenous ketone supplementation on skeletal muscle metabolism and exercise. The authors reviewed the literature widely, then I learned a lot from this manuscript. However, in some parts discussing influences of ketogenic diet consumption, whether those effects were due to ketone body per se or other factors induced by high fat exposure and/or low carbohydrate level were not sufficiently organized in the manuscript. I think this manuscript would become better if this point was improved.
We thank the reviewer for their feedback on the review. The question about whether the effects of a keto diet are due to ketones themselves or the high fat nature of the diet is important. However, unfortunately, based on the design of most existing studies, it is not possible to make this distinction (e.g., they do not implement both a subject group on a keto diet and a subject group where ketolysis or exposure to ketones is increased independently of increased fat administration). This is a question that should be addressed going forward. We have added a comment starting on line 623 that highlights this, and have made hypothesis or suggestions on this subject where possible (line 173, 414, 458, 522 and 542).
- Effects of exercise on muscle ketone body metabolism
All studies referred in this section were performed in animals or cells and it seems that no investigation related to the effects of ketone bodies on enzymatic activities has been performed in human subjects. I recommend mentioning it as a limitation.
Thank you for mentioning this observation. We have added a sentence on line 152 to address this comment.
- Muscle PDK4 activation following a keto diet and exercise.
Please provide background information on why you focus on the regulation of pyruvate dehydrogenase regulation in this review. How did you hypothesize that ketone bodies influence PDH regulation?
This is a good question. We did not begin working on this review with a hypothesis that ketone bodies influence PDH regulation. The section on PDK4 was created because many studies we came across in a general literature search on muscle + ketones + exercise measured the levels of PDK4. Since PDK4 is linked to glucose metabolism, and seems to be activated to spare glucose levels when glycogen is low, it became apparent that it is likely relevant to the low carb nature of a keto diet. Therefore, we decided to add a section on PDK4. We modified the introduction to the section (see line 163-164) in order to address this comment.
- Alterations in AMPK and PCG1α activation following an exercise and keto regimen
According to the second paragraph, it seems that the upregulation of PGC1a expression in skeletal muscle induced by a ketogenic diet was due to high fat exposure and low glycogen level. Is there any study that examined the effects of ketone bodies per se on PGC-1a expression in skeletal muscle (or C2C12 cells)?
We thank the reviewer for this insightful remark. Unfortunately, we were unable to find any studies that investigated whether ketone bodies themselves can influence PGC1a expression. However, we agree with the reviewer that this would be a beneficial question to answer in order to separate the high fat diet influence of a keto diet from the effect of the ketone bodies themselves. To this point, we have added a sentence starting on line 414.
- Influence of increased exposure to ketones on glucose metabolism
A ketogenic diet is a very low carbohydrate diet. It has been suggested that the percentage of carbohydrates in diet would influence skeletal muscle GLUT4 expression (for example, PMID: 8419118, 11473054, 33456007). Moreover, it is well known that high fat (low carbohydrate) diet induces insulin resistance. I recommend that the authors reconstitute this section to organize the effects of ketone bodies per se and a ketogenic diet (= very low carbohydrate diet) on glucose metabolism in skeletal muscle. Many recent studies have shown that ketone monoester intake attenuated the increase in blood glucose levels in OGTT tests (e.g., PMID: 33010116, 31599919, 32941737). Some of these reported that ketone monoester intake did not influence plasma insulin level or gastric emptying, suggesting that ketone body would impact on peripheral insulin action. If appropriate, I recommend discussing these studies in this section.
We greatly appreciate that the reviewer found and shared specific PMID numbers with us – the articles are very relevant and thought-provoking. We have incorporated a selection of these articles into the review, which can be found on lines 458-467, 474-481 and mentioned on line 629.
Thank you for the suggestion to separate the section into the effect of keto diets and ketone bodies themselves. Where possible, we organized other sections in the review this way because we agree with the reviewer that it helps draw comparisons for the reader. We have reorganized this section accordingly.
Lines 456-460 Please show the references of these two sentences.
Thank you for noticing that the reference was missing. Reference 49 that was previously shown at the end of this paragraph was actually the reference for these 2 sentences. We moved it to line 470 to address this comment. Note that the reference is now number 52.
- Impact of ketones on muscle health following exercise
In this section, I recommend reviewing reference 32 and the subsequent study of that (PMID: 37062892). Reference 32
investigated effects of ketone monoester intake after the daily training and they showed the group who took ketone monoester after exercise could train with a greater work output. The same group showed in PMID: 37062892 that this might stimulate exercise-induced adaptation in skeletal muscle.
Thank you for pointing this out and sharing the additional article with us. Instead of adding the additional information from reference 32 to section 10, we chose to incorporate it into section 5 because it fit nicely with the discussion on exercise performance (addition starts on line 341). We then mentioned that the articles are connected in section 10 and we did incorporate the subsequent study in the muscle health section (addition starts on line 587).

Reviewer 2 Report
Dear authors;
I've had the opportunity to review your manuscript, and I must commend you on a well-crafted and exhaustive review of the topic. Your efforts in collating and presenting the information are evident, and the depth of your analysis is commendable.
However, while the foundation of your review is strong, I believe that it would benefit from major revisions to further enhance its quality and utility for readers. Incorporating these changes would undoubtedly elevate the impact and comprehensiveness of your work.
1. The abstract mentions the importance of skeletal muscle twice (lines 15 and 18). This can be condensed to avoid redundancy.
2. The production of ketone bodies during fasting and their role as an alternative energy source is mentioned twice (lines 27-28 and 12-13). This can be streamlined.
3. Lines 68-70: The introduction to this section is a bit repetitive. The phrase "effect of exercise and keto diets or exogenous ketone supplementation on parameters linked to ketone body metabolism" is quite long and could be streamlined for clarity.
4. Lines 76-80: The mention of the same mice and the effects observed on them could be more concise. The current structure might confuse readers about which effects were observed under which conditions.
5. Lines 101-109: The discussion about the impact of exercise on enzyme activity is spread out. It might be clearer to group similar findings together and differentiate between different muscles or muscle types more distinctly.
6. 576-584: Emphasize the importance of diverse study designs and participant profiles for a more comprehensive understanding of the topic.
7. 585-588: The need for more inclusive research (including females) is crucial and should be emphasized more strongly.
8. 588: A call to action for more research in this area could be a strong ending to the conclusion.
9. Review Limitations: It's essential to highlight the limitations of the review. This could encompass:
· A lack of studies in certain areas or subtopics.
· Variability in study designs, methodologies, or study populations.
· The potential for publication bias (i.e., the tendency to publish positive results over negative ones).
10. Future Research Directions: While the article does touch upon the need for more research in certain areas, it would be beneficial to have a dedicated section that:
· Proposes specific studies or experimental designs.
· Suggests research in areas that have yet to be explored.
· Recommends methodologies or approaches that could address current limitations.
11. Inclusivity in Research: The article points out the lack of studies involving female participants or female animals. This is a significant gap in the literature and could be a key area for future research.
12. Practical Implications: Even for a review article, readers often look for practical implications or recommendations based on the reviewed literature. A section highlighting how the findings can be applied in practice (e.g., sports training, nutrition, or medicine) would be valuable.
13. Comparison with Other Reviews: If there are other reviews on the same topic, it would be beneficial to compare and contrast the findings of this article with those works to provide a more comprehensive perspective.
14. Summary Table: Above all, a summary table would be an outstanding addition to offer a quick and clear view of the existing literature. This table could include:
· Study reference.
· Study design (e.g., duration, type of exercise, keto diet composition).
· Study population (e.g., gender, age, activity level).
· Key findings or outcomes.
· Study limitations.
By incorporating these elements, the review would be enriched, offering a more complete and useful resource for readers.
Author Response
Response to R2
Dear authors;
I've had the opportunity to review your manuscript, and I must commend you on a well-crafted and exhaustive review of the topic. Your efforts in collating and presenting the information are evident, and the depth of your analysis is commendable.
However, while the foundation of your review is strong, I believe that it would benefit from major revisions to further enhance its quality and utility for readers. Incorporating these changes would undoubtedly elevate the impact and comprehensiveness of your work.
We thank the reviewer for their feedback on the review.
- The abstract mentions the importance of skeletal muscle twice (lines 15 and 18). This can be condensed to avoid redundancy.
Thank you for this suggestion, but we felt that it was important to mention the skeletal muscle in both sentences. Line 15 (now line 14) was added to explain why the skeletal muscle is important, while line 18 (part of the sentence explaining the goal of the review) is meant to emphasize that this review focuses on the skeletal muscle and not other tissues.
- The production of ketone bodies during fasting and their role as an alternative energy source is mentioned twice (lines 27-28 and 12-13). This can be streamlined.
This comment was removed from the start of the abstract (line 12). Thank you for pointing this out.
- Lines 68-70: The introduction to this section is a bit repetitive. The phrase "effect of exercise and keto diets or exogenous ketone supplementation on parameters linked to ketone body metabolism" is quite long and could be streamlined for clarity.
Thank you for the suggestion. We have modified the introduction to this section to make it clearer. Note that the relevant text is now on lines 67-69.
- Lines 76-80: The mention of the same mice and the effects observed on them could be more concise. The current structure might confuse readers about which effects were observed under which conditions.
Thank you for this comment. We have broken up the sentences in order to make the information easier to understand.
- Lines 101-109: The discussion about the impact of exercise on enzyme activity is spread out. It might be clearer to group similar findings together and differentiate between different muscles or muscle types more distinctly.
We appreciate the suggestion from the reviewer. These lines were organized by muscle type (gastrocnemius, then respiratory muscles). We have now specified the type of muscle (glycolytic or mixed) and added a hypothesis that the effects may not be seen in glycolytic muscle. See these additions spread over lines 102-113.
- 576-584: Emphasize the importance of diverse study designs and participant profiles for a more comprehensive understanding of the topic.
We agree with the reviewer and appreciate their suggestion. Lines 639-642 were added to address this comment.
- 585-588: The need for more inclusive research (including females) is crucial and should be emphasized more strongly.
We also agree with the reviewer with respect to this comment. Lines 645-647 were added to highlight the need for more inclusive research.
- 588: A call to action for more research in this area could be a strong ending to the conclusion.
Thank you for the suggestion, we agree that this is an impactful way to finish the review. Lines 660-663 were added to address this comment.
- Review Limitations: It's essential to highlight the limitations of the review. This could encompass:
- A lack of studies in certain areas or subtopics.
- Variability in study designs, methodologies, or study populations.
- The potential for publication bias (i.e., the tendency to publish positive results over negative ones)
See response below in #10.
- Future Research Directions: While the article does touch upon the need for more research in certain areas, it would be beneficial to have a dedicated section that:
- Proposes specific studies or experimental designs.
- Suggests research in areas that have yet to be explored.
- Recommends methodologies or approaches that could address current limitations.
Thank you for both of these suggestions (#9 and 10). Section 11 in the first draft has been split into 3 new sections (11 – summary; 12 – limitations and future direction; 13- conclusion). We felt that since limitations lead directly into future avenues for research, that these sections could be combined to avoid repetition. Therefore, section 12 summarizes limitations of the review and has recommended directions for future research. We agree with the reviewer that making more distinct sections assists in highlighting the different topics to the reader. Note that new text is highlighted, but the arrangement of some pre-existing text may have been modified to fit the new sections.
- Inclusivity in Research: The article points out the lack of studies involving female participants or female animals. This is a significant gap in the literature and could be a key area for future research.
We agree with the reviewer that the lack of female participants is a detrimental gap in the literature. Lines 643-647 were modified/added to highlight the need to integrate and compare both biological sexes.
- Practical Implications: Even for a review article, readers often look for practical implications or recommendations based on the reviewed literature. A section highlighting how the findings can be applied in practice (e.g., sports training, nutrition, or medicine) would be valuable.
Thank you for the suggestion, we agree that providing this type context for the reader is beneficial. We attempted to address this comment on lines 655-660.
- Comparison with Other Reviews: If there are other reviews on the same topic, it would be beneficial to compare and contrast the findings of this article with those works to provide a more comprehensive perspective.
This is an interesting suggestion. Unfortunately, we are only able to find review articles that summarize the literature on ketones and exercise performance. The goal of this review was to highlight the effect of ketones and exercise specifically on muscle metabolism, and we were not able to find any existing review articles that touch on this specific subject.
- Summary Table: Above all, a summary table would be an outstanding addition to offer a quick and clear view of the existing literature. This table could include:
- Study reference.
- Study design (e.g., duration, type of exercise, keto diet composition).
- Study population (e.g., gender, age, activity level).
- Key findings or outcomes.
- Study limitations.
We thank the reviewer for this suggestion. A table has been created and incorporated into the review.
By incorporating these elements, the review would be enriched, offering a more complete and useful resource for readers.

Reviewer 3 Report
The manuscript under revision provides a comprehensive depth and a balanced perspective on the current knowledge, challenges, research gaps and future developments regarding the relationship between ketone body metabolism and exercise in the skeletal muscle.
Line 60: references are missing
Line 578: may be helpful for the reader to specifically comment, in short, the positive and negative influence of ketones on muscle metabolism
Figure 1: references are required for all effects depicted the cassettes (↑,↓,x); the significance of the color code used for the symbols should be provided (red, green, violet, grey; why the grey ones do not have references?)
A table with beneficial/detrimental effects of ketone bodies (diet/exogenous) in combination with physical activity on skeletal muscle metabolism would be welcomed
Author Response
Response to R3
The manuscript under revision provides a comprehensive depth and a balanced perspective on the current knowledge, challenges, research gaps and future developments regarding the relationship between ketone body metabolism and exercise in the skeletal muscle.
We thank the reviewer for their feedback on the review.
Line 60: references are missing
Thank you for pointing this out. The sentence ending on line 60 (now line 58) refers to all the articles summarized in this review. Line 58 was modified to make this more apparent.
Line 578: may be helpful for the reader to specifically comment, in short, the positive and negative influence of ketones on muscle metabolism
We thank the reviewer for this suggestion. This comment was addressed in lines 597-616. However, we did rearrange the previous conclusion into more distinct sections, in part to make the answer to this comment more apparent.
Figure 1: references are required for all effects depicted the cassettes (↑,↓,x); the significance of the color code used for the symbols should be provided (red, green, violet, grey; why the grey ones do not have references?)
We thank the reviewer for their feedback. We intended the grey symbols to indicate that there were no pertinent references (for example, we found no articles showing that a keto diet induced decreased myokine secretion, i.e., we left a grey ↑ in that section). This is something we should have explained. However, we acknowledge that the grey symbols cause unnecessary confusion, so we have removed the grey symbols from the sections that do not have pertinent references. The color code for the symbols was chosen arbitrarily for visual aesthetic – the figure caption has been updated to associate the different symbols with their colors (e.g., purple X).
A table with beneficial/detrimental effects of ketone bodies (diet/exogenous) in combination with physical activity on skeletal muscle metabolism would be welcomed
We thank the reviewer for this suggestion. A table has been created and incorporated into the review.

Round 2
Reviewer 1 Report
Overall points.
The authors have sufficiently addressed my concerns, and the revised version of the manuscript has significantly improved from its original version. However, there are some minor points that I would like to improve.
3. Muscle PDK4 activation following a keto diet and exercise.
Lines 163-164 I think “a lack of glucose available to build muscle glycogen” would not directly activate PDK4. There should be another possible mechanism related to the keto (very low carbohydrate and high fat) diet that influences PDK4 activity or PDK4 expression.
7. Influence of increased exposure to ketones on glucose metabolism
Line 462 “the low carbohydrate content of keto diets can influence GLUT4 protein level.”
Lines 464-467 Previous two sentences discussed the influences of the keto diet on GLUT4 expression. But this sentence is not relevant to that. I recommend moving this sentence to line 461.
Author Response
Thank you for your comments. We have answered them in the attached document.
- Muscle PDK4 activation following a keto diet and exercise.
Lines 163-164 I think “a lack of glucose available to build muscle glycogen” would not directly activate PDK4. There should be another possible mechanism related to the keto (very low carbohydrate and high fat) diet that influences PDK4 activity or PDK4 expression.
We thank the reviewer for this suggestion. We had discussed the possibility that the high fat diet nature of the keto diet could also modify PDK4 at the end of the paragraph. So, we moved this comment next to the hypothesis about glycogen on lines 163-164
- Influence of increased exposure to ketones on glucose metabolism
Line 462 “the low carbohydrate content of keto diets can influence GLUT4 protein level.”
Lines 464-467 Previous two sentences discussed the influences of the keto diet on GLUT4 expression. But this sentence is not relevant to that. I recommend moving this sentence to line 461.
Thank you for this suggestion. We believe that lines 464-467 should remain where they are, as the article is referring to a high carbohydrate diet, not a keto diet. We put the article in this place to support our discussion on lines 458-467 about the effect of the low carbohydrate component of the keto diet. We did modify line 462 to make this more clear.

Reviewer 2 Report
The authors have appropriately addressed all the raised issues. Congratulations.
Author Response
Thank you for having helped us to improve our review.